# Quasinormal Modes of a Charged Black Hole with Scalar Hair

**Wen-Di Guo** [1,2,*,†] and **Qin Tan** [1,2,†]

1    Lanzhou Center for Theoretical Physics, Key Laboratory of Theoretical Physics of Gansu Province, School of Physical Science and Technology, Lanzhou University, Lanzhou 730000, China; tanq19@lzu.edu.cn
2    Institute of Theoretical Physics, Research Center of Gravitation, Lanzhou University, Lanzhou 730000, China
*    Correspondence: guowd@lzu.edu.cn
†    These authors contributed equally to this work.

**Abstract:** Based on the five-dimensional Einstein–Maxwell theory, Bah et al. constructed a singularity-free topology star/black hole [Phys. Rev. Lett. 126, 151101 (2021)]. After performing the Kaluza–Klein reduction, i.e., integrating the extra space dimension, it can obtain an effective four-dimensional spherically static charged black hole with scalar hair. In this paper, we study the quasinormal modes (QNMs) of the scalar, electromagnetic, and gravitational fields in the background of this effective four-dimensional charged black hole. The radial parts of the perturbed fields all satisfy a Schrödinger-like equation. Using the asymptotic iteration method, we obtain the QNM frequencies semianalytically. For low-overtone QNMs, the results obtained using both the asymptotic iteration method and the Wentzel–Kramers–Brillouin approximation method agree well. In the null coordinates, the evolution of a Gaussian package is also studied. The QNM frequencies obtained by fitting the evolution data also agree well with the results obtained using the asymptotic iteration method.

**Keywords:** magnetic charged black hole; quasinormal modes; gravitational waves

**PACS:** 04.70.-s; 04.70.Bw

## 1. Introduction

In 2015, the Laser Interferometer Gravitational-Wave Observatory (LIGO) and Virgo detected the gravitational wave of a binary black hole system [1], which opened the window to gravitational wave astrophysics. Subsequently, the Event Horizon Telescope (EHT) took the first pictures of the supermassive black hole at the center of Galaxy M87 in 2019 [2–7] and the black hole in our Milky Way in 2022 [8–13]. This enhanced our ability to test fundamental physical problems. One is whether a singularity exists [14,15]. The search for black hole alternatives has attracted a lot of interest. Ultra-compact objects, such as gravastars [16], boson stars [17], and wormholes [18–21], have been constructed. More details can be found in the review by [22] and the references therein. However, these models usually need some exotic matter and their UV origin is not clear. On the other hand, from string theory, which is the most important candidate for unifying quantum theory and gravity theory, some horizonless models have been constructed. One of these, a fuzzball, has a smooth microstate geometry and is similar to classical black holes up to the Planck scale [23]. However, constructing a fuzzball requires many degrees of freedom, and it is difficult to study astrophysical observations [24–26]. To address these disadvantages, Bah and Heidmann proposed a topological star/black hole model, which can be constructed from type-II B string theory and is similar to classical black holes in macrostate geometries [27,28]. So, studying astrophysical observations is not too challenging. They further carefully analyzed the thermodynamic stability of the solution [29]. In addition, the motion of a charged particle on the background of this topological star/black hole model has been studied [30]. Based on this solution, a four-dimensional spherically static charged black hole with scalar hair could be obtained by

integrating an extra dimension [27,28]. We will study the quasinormal modes (QNMs) of the scalar, electromagnetic, and gravitational fields in the background of the black hole in this paper.

Quasinormal modes are characteristic of dissipative systems and play important roles in various physical areas. Classically, everything that falls into the interior of the black hole cannot escape because of the event horizon; therefore, black holes are dissipative systems. The quasinormal modes of black holes dominate the ringdown stage, which is the final stage of gravitational waves for a binary black hole merging system [31]. A big difference between QNMs and normal modes is that the eigenfunctions of normal modes form a complete set, whereas those of QNMs do not [32]. Moreover, the eigenfunctions of QNMs are not normalizable [32]. Quasinormal modes have complex frequencies, where the real parts are the vibration frequencies and the imaginary parts are the inverse of the decay time scale of the perturbation. Therefore, it is very important to study the QNMs of black holes. The mass and angular momentum can be inferred from QNMs, and the no-hair theorem can also be tested using QNMs [33–35]. For horizonless compact objects, there could be echoes in the ringdown signal, which is the smoking gun for the existence of the horizonless compact objects [15,22,36]. One can also use the QNMs to constrain modified gravity theories [37–45]. It has also been found that the QNM spectrum is unstable under a small perturbation of the potential [46,47]. Furthermore, the QNM frequencies can also partly reveal the stability of the background space-time under small perturbations [48,49]. QNMs play very important roles in other physical systems, for example, leaky resonant cavities [50], and brane world models [51–53], which have been widely studied [54–62].

In this paper, we will study the QNMs in the background of a four-dimensional spherically static black hole with scalar hair. The remainder of this paper is organized as follows. In Section 2, we provide a brief overview of the charged black hole with scalar hair, as well as the KK reduction. In Section 3, we study small perturbations in the scalar, electromagnetic, and gravitational fields. By expanding the perturbation fields in the spherical harmonic function, we can derive the master equations of the perturbations. In Section 4, we compute the QNM frequencies using the asymptotic iteration method (AIM) and Wentzel–Kramers–Brillouin (WKB) approximation method, as well as the time evolution of a Gaussian package. Finally, we present the conclusions in Section 5.

## 2. The Charged Black Hole

The nonsingular black hole/topology star proposed by I. Bah and P. Hedmann [27,28] originates from the action of a five-dimensional Einstein–Maxwell theory

$$S = \int d^5 x \sqrt{-\hat{g}} \left( \frac{1}{2\kappa_5^2} \hat{R} - \frac{1}{4} \hat{F}^{MN} \hat{F}_{MN} \right), \tag{1}$$

where $\hat{F}_{MN}$ is the five-dimensional electromagnetic field tensor, and $\kappa_5$ is the five-dimensional gravitational constant. We use the hat symbol to denote the five-dimensional quantities. The five-dimensional coordinates are denoted by $M, N \dots$. The metric is considered spherically symmetric [63].

$$
\begin{aligned}
ds^2 &= -f_S(r)dt^2 + f_B(r)dy^2 + \frac{1}{f_S(r)f_B(r)}dr^2 \\
&+ r^2 d\theta^2 + r^2 \sin^2\theta d\phi^2.
\end{aligned} \tag{2}
$$

The extra dimension coordinate is denoted by $y$. The field strength of the magnetic field is

$$\hat{F} = P \sin\theta d\theta \wedge d\phi. \tag{3}$$

The solution can be solved according to [63].

$$f_B(r) = 1 - \frac{r_B}{r}, \qquad f_S(r) = 1 - \frac{r_S}{r}, \qquad P = \pm \frac{1}{\kappa_5^2} \sqrt{\frac{3 r_S r_B}{2}}. \tag{4}$$

Metric (2) is symmetric under a double rotation, which transforms the coordinate ($t$, $y$, $r_S$, $r_B$) to ($iy$, $it$, $r_B$, $r_S$). At $r = r_S$, the space-time features a horizon, where $f_S(r) = 0$, whereas at $r = r_B$, a degeneracy of the $y$-circle occurs. Bah and Heidmann showed that there are smooth bubbles at $r = r_B$, which terminate the space-time [27,28]. The space-time has two configurations. One is a black string, where $r_S \geq r_B$ because the bubble hides behind the horizon, and the other is a topology star, where the horizon disappears when $r_S < r_B$ because the space-time terminates at $r = r_B$ [27,28].

We rewrite the metric (2) as

$$ds_5^2 = e^{2\Phi} ds_4^2 + e^{-4\Phi} dy^2, \tag{5}$$

$$ds_4^2 = f_B^{\frac{1}{2}} \left( -f_S dt^2 + \frac{dr^2}{f_B f_S} + r^2 d\theta^2 + r^2 \sin^2 \theta d\phi^2 \right), \tag{6}$$

where

$$e^{2\Phi} = f_B^{-1/2}, \tag{7}$$

and $\Phi$ is a dilaton field. After performing the Kaluza–Klein reduction, i.e., integrating the extra dimension, we can obtain a four-dimensional theory known as the Einstein–Maxwell dilaton theory

$$\begin{aligned} S_4 = & \int d^4 x \sqrt{-g} \left( \frac{1}{2\kappa_4^2} R_4 - \frac{3}{\kappa_4^2} g^{\mu\nu} \partial_\mu \Phi \partial_\nu \Phi \right. \\ & \left. - 2\pi R_y e^{-2\Phi} F_{\mu\nu} F^{\mu\nu} \right), \end{aligned} \tag{8}$$

where $R_y$ is the radius of the extra dimension. We use $\mu, \nu \ldots$ to label the four-dimensional coordinates. Here, the quantities without the hat symbol are constructed in the four-dimensional space-time. The four-dimensional gravitational constant is defined as

$$\kappa_4 = \frac{\kappa_5}{\sqrt{2\pi R_y}}. \tag{9}$$

We can solve the four-dimensional field strength of the magnetic field as follows:

$$F = \pm \frac{1}{\kappa_4 \sqrt{2\pi R_y}} \sqrt{\frac{3 r_B r_S}{2}} \sin \theta d\theta \wedge d\phi. \tag{10}$$

The ADM mass $M$ and magnetic charge $Q_m$ can be solved as follows:

$$\begin{aligned} M &= 2\pi \left( \frac{2 r_S + r_B}{\kappa_4^2} \right), \\ Q_m &= \frac{1}{\kappa_4} \sqrt{\frac{3}{2} r_B r_S}. \end{aligned} \tag{11}$$

Or, in terms of $M$ and $Q_m$,

$$r_S^{(1)} = \frac{\kappa_4^2}{8\pi} (M - M_\triangle), \qquad r_B^{(1)} = \frac{\kappa_4^2}{4\pi} (M + M_\triangle), \tag{12}$$

$$r_S^{(2)} = \frac{\kappa_4^2}{8\pi} (M + M_\triangle), \qquad r_B^{(2)} = \frac{\kappa_4^2}{4\pi} (M - M_\triangle), \tag{13}$$

where $M_\triangle^2 = M^2 - \left(\frac{8\pi Q_{\rm m}}{\sqrt{3}\kappa_4}\right)^2$. From Solution (4), we know that when $r < r_B$, $f_B^{1/2}$ becomes imaginary, and the metric is unphysical. So, the coordinate $r$ cannot be smaller than $r_B$. Moreover, the black string scenario has Gregory–Laflamme instability [64]. However, models with compact extra dimensions that result in a discrete KK mass spectrum can potentially avoid this instability. Stotyn and Mann have demonstrated that when $R_y > \frac{4\sqrt{3}}{3}Q_{\rm m}$, Solution (13) is stable. It is important to note that the space-time at $r = r_B$ is singular. Thus, when $r_S \geq r_B$, Metric (6) describes a charged black hole with scalar hair. We only study this case in this paper. it is important to note that magnetically charged black holes have been studied in [65–67].

### 3. Perturbation Equations

In this section, we analyze the linear perturbation equations of the scalar, electromagnetic, and gravitational fields in the background of a charged black hole with scalar hair. For simplicity, we consider the three types of perturbed fields separately, that is, when one field is perturbed, the background of the other two fields is unaffected. However, it is important to note that perturbations of the electromagnetic and gravitational fields are typically coupled together for charged black holes.

*3.1. Scalar Field*

We consider a free massless scalar field in the background of this charged black hole. The equation of motion for the scalar field is the Klein–Gordon equation

$$\frac{1}{\sqrt{-g}}\partial_\mu\left(\sqrt{-g}\partial^\mu\varphi\right) = 0. \tag{14}$$

Because of the spherical symmetry and time independence of the background, we can decompose the scalar field as follows:

$$\varphi(t,r,\theta,\phi) = \sum_{l,m} e^{-i\omega t} f_B^{-1/4}\frac{1}{r}\psi_{\rm s}(r)Y_{l,m}(\theta,\phi), \tag{15}$$

where $Y_{l,m}$ is the spherical harmonics that satisfies

$$\triangle Y_{l,m} = -l(l+1)Y_{l,m}, \tag{16}$$

where $\triangle$ is the Laplace–Beltrami operator. By substituting this into Equation (14), we obtain the radial part of the perturbation equation for the scalar field

$$f_S^2 f_B\psi_{\rm s}'' + f_S(f_S'f_B + \frac{1}{2}f_B'f_S)\psi_{\rm s}' + (\omega^2 - V_{\rm s}(r))\psi_{\rm s} = 0, \tag{17}$$

where

$$V_{\rm s}(r) = f_S\left(\frac{l(l+1)}{r^2} + \frac{1}{4}f_Sf_B'' + f_B'f_S' - \frac{f_Sf_B'^2}{4f_B}\right)$$
$$+ \frac{f_S}{r}(f_Sf_B' + f_Bf_S') \tag{18}$$

is the effective potential for the scalar field. Hereafter, we use a prime to denote the derivative with respect to the coordinate $r$. In order to obtain the Schrödinger-like equation, we require the tortoise coordinate $r_*$, which can be obtained from the following relation:

$$dr_* = \frac{1}{\sqrt{f_B}f_S}dr. \tag{19}$$

In this way, Equation (17) can be written as

$$\frac{d^2\psi(r_*)}{dr_*^2} + (\omega^2 - V_s(r_*))\psi(r_*) = 0.$$ (20)

*3.2. Electromagnetic Field*

For an electromagnetic field, the Maxwell equation is given by

$$\frac{1}{\sqrt{-g}}\partial_\mu\left(\sqrt{-g}\tilde{F}^{\mu\nu}\right) = 0,$$ (21)

where $\tilde{F}_{\mu\nu} = \partial_\mu\tilde{A}_\nu - \partial_\nu\tilde{A}_\mu$ is the field-strength tensor of the perturbed electromagnetic field $\tilde{A}_\mu$. To separate the perturbed electromagnetic field, we require the vectorial spherical harmonics, which are defined as [68–70]

$$(V_{l,m}^1)_a = \partial_a Y_{l,m}(\theta,\phi),$$ (22)

$$(V_{l,m}^2)_a = \gamma^{bc}\epsilon_{ac}\partial_b Y_{l,m}(\theta,\phi).$$ (23)

Here, $a, b, c$ denote the angular coordinates $\theta$ and $\phi$. $\gamma$ refers to the induced metric on the sphere with radius 1, and $\epsilon$ is the totally antisymmetric tensor in two dimensions. It is important to note that $(V_{l,m}^1)_a$ and $(V_{l,m}^2)_a$ behave differently under space inversion, i.e., $(\theta,\phi) \rightarrow (\pi - \theta, \pi + \phi)$. $(V_{l,m}^1)_a$ is even or polar, that is, it acquires a factor of $(-1)^l$ under space inversion. $(V_{l,m}^2)_a$ is odd or axial, that is, it acquires a factor of $(-1)^{l+1}$ under space inversion. Thus, the perturbed electromagnetic field $\tilde{A}_\mu$ can be decomposed as

$$\tilde{A}_\mu(t,r,\theta,\phi) = \sum_{l,m} e^{-i\omega t}\begin{bmatrix} 0 \\ 0 \\ \frac{\psi_v(r)}{\sin\theta}\frac{\partial Y_{l,m}}{\partial\phi} \\ -\psi_v(r)\sin\theta\frac{\partial Y_{l,m}}{\partial\theta} \end{bmatrix}$$

$$+ \sum_{l,m} e^{-i\omega t}\begin{bmatrix} h_1(r)Y_{l,,m} \\ h_2(r)Y_{l,,m} \\ h_3(r)\frac{\partial Y_{l,m}}{\partial\theta} \\ h_3(r)\frac{\partial Y_{l,m}}{\partial\phi} \end{bmatrix}.$$ (24)

Owing to the spherical symmetry of the background metric, the perturbation equations do not mix polar and axial contributions. Moreover, the axial and polar parts contribute equally to the final result [68,69]. So, we only need to deal with the axial part. By substituting the background metric from (6) into the Maxwell equation in (21), we obtain the perturbation equation for the radial part $\psi_v$

$$f_S^2 f_B \psi_v'' + f_S(f_S' f_B + \frac{1}{2}f_B' f_S)\psi_v' + (\omega^2 - V_v(r))\psi_v = 0,$$ (25)

where the effective potential is

$$V_v(r) = \frac{f_S(r)l(l+1)}{r^2},$$ (26)

which does not depend on the parameter $r_B$. However, the effective potential depends on the parameter $r_S$, which is related to the magnetic charge $Q_m$. Using the tortoise coordinate $r_*$, the perturbation equation can also be transformed into a Schrödinger-like form. From Equation (19), we know that the tortoise coordinate $r_*$ depends on the parameter $r_B$ so the QNMs will also be affected by the parameter $r_B$.

*3.3. Gravitational Field*

　　Considering a perturbation on the background metric (6), the perturbed metric is

$$\bar{g}_{\mu\nu} = g_{\mu\nu} + h_{\mu\nu}, \tag{27}$$

where $h_{\mu\nu}$ is the perturbation. The separation of the perturbation for the gravitational field is more complicated. In addition to the vectorial spherical harmonics, we also need to consider the tensorial harmonics, which are defined as [71]

$$(T_{l,m}^1)_{ab} = (Y_{l,m})_{;ab}, \tag{28}$$

$$(T_{l,m}^2)_{ab} = Y_{l,m}\gamma_{ab}, \tag{29}$$

$$(T_{l,m}^3)_{ab} = \frac{1}{2}[\epsilon_a^c(Y_{l,m})_{;cb} + \epsilon_b^c(Y_{l,m})_{;ca}], \tag{30}$$

where the semicolon denotes the covariant derivative on the sphere. Among them, $T_{l,m}^3$ is odd under space inversion, whereas the other two are even. Based on the principle of general covariance, the theory should maintain covariance under an infinitesimal coordinate transformation. Thus, we can choose a specific gauge to simplify the problem. In the Regge–Wheeler gauge [71], the perturbation $h_{\mu\nu}$ can be written as

$$h_{\mu\nu} = \sum_l e^{-i\omega t} \begin{bmatrix} 0 & 0 & 0 & h_0(r) \\ 0 & 0 & 0 & h_1(r) \\ 0 & 0 & 0 & 0 \\ h_0(r) & h_1(r) & 0 & 0 \end{bmatrix} \sin\theta\partial_\theta Y_{l,0}(\theta) \tag{31}$$

for the odd parity, and

$$h_{\mu\nu} = \sum_l e^{-i\omega t} \begin{bmatrix} H_0(r) & H_1(r) & 0 & 0 \\ H_1(r) & H_2(r) & 0 & 0 \\ 0 & 0 & r^2 K(r) & 0 \\ 0 & 0 & 0 & r^2 K(r)\sin^2\theta \end{bmatrix} Y_{l,0}(\theta) \tag{32}$$

for the even parity. For simplicity, we have chosen $m = 0$ because the perturbation equations do not depend on the value of $m$ [71]. For the Schwarzschild black hole, the odd and even parities have the same QNM spectrum [72], whereas other black holes may not possess this property. In this paper, we focus on the odd parity for simplicity. By substituting the decomposition into Einstein's equation and performing some algebraic operations, the perturbation equations for the odd parity perturbation can be combined into a single equation for the variable $\psi_g$, which is defined as

$$\psi_g(r) = \frac{f_B^{1/4}(r)f_S(r)}{r}h_1(r). \tag{33}$$

　　The variable $\psi_g$ satisfies the Schrödinger-like Equation (20), with the effective potential

$$V_g(r) = f_S\left(\frac{3}{4}f_B'f_S' + f_Bf_S'' - \frac{f_Bf_S'}{r} + \frac{l(l+1)}{r^2}\right)$$
$$- f_S^2\left(\frac{9f_B'^2}{16f_B} - \frac{3f_B''}{4}\right). \tag{34}$$

　　The effective potential plays an important role in determining the value of the QNMs. We plot the effective potentials for the scalar, electromagnetic, and gravitational fields in Figure 1. All of the effective potentials approach zero at the horizon and infinity, which is similar to the Schwarzschild black hole. In addition, the effective potentials for the scalar and gravitational fields depend on the parameter $r_B$.

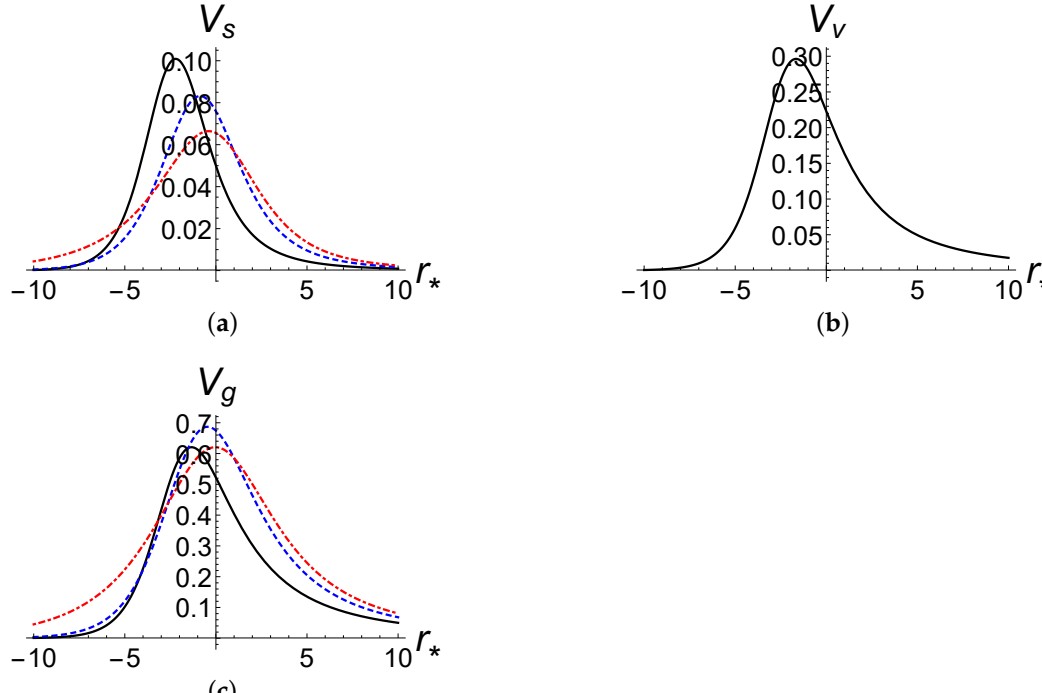

**Figure 1.** The effective potentials in the tortoise coordinate $r_*$. The parameter $r_B$ is set to $r_B = 0.1 r_S$ (black solid lines), $r_B = 0.5 r_S$ (blue dashed lines), and $r_B = 0.9 r_S$ (red dashed lines). (**a**) The scalar field with $l = 0$. (**b**) The electromagnetic field with $l = 1$. (**c**) The gravitational field with $l = 2$.

## 4. Computing QNMs

In this section, we present the solutions for the QNMs of the charged black hole with a scalar field, considering both the frequency and time domains. Additionally, we compare the results using two different methods.

### 4.1. Solving Frequency

First, we compute the frequencies of the QNMs for the three types of fields using the AIM and WKB approximation method. Here, we provide a brief overview of the AIM.

Consider a homogeneous linear second-order differential equation given by

$$\chi''(x) = \lambda_0(x)\chi'(x) + s_0(x)\chi(x), \tag{35}$$

where $\lambda_0(x)$ and $s_0(x)$ are smooth functions. Based on the symmetric structure of the right-hand side of Equation (35), we can find a general solution to this equation [73]. By differentiating Equation (35) with respect to the variable $x$, we obtain

$$\chi'''(x) = \lambda_1(x)\chi'(x) + s_1(x)\chi(x), \tag{36}$$

where

$$\begin{aligned} \lambda_1(x) &= \lambda_0'(x) + s_0(x) + (\lambda_0)^2, \\ s_1(x) &= s_0'(x) + s_0(x)\lambda_0(x). \end{aligned} \tag{37}$$

By differentiating Equation (36) with respect to $x$, we find that

$$\chi''''(x) = \lambda_2(x)\chi'(x) + s_2(x)\chi(x), \tag{38}$$

where

$$\begin{aligned}
\lambda_2(x) &= \lambda_1'(x) + s_1(x) + \lambda_0\lambda_1, \\
s_2(x) &= s_0'(x) + s_0(x)\lambda_1(x).
\end{aligned} \tag{39}$$

By continuing this process, the $(n+1)$-th and $(n+2)$-th derivatives give us the following relations

$$\begin{aligned}
\lambda_n(x) &= \lambda_{n-1}'(x) + s_{n-1}(x) + \lambda_0(x)\lambda_{n-1}(x), \\
s_n(x) &= s_{n-1}'(x) + s_0(x)\lambda_{n-1}(x).
\end{aligned} \tag{40}$$

When $n$ is sufficiently large, the asymptotic aspect can be introduced as

$$\frac{s_n(x)}{\lambda_n(x)} = \frac{s_{n-1}(x)}{\lambda_{n-1}(x)}. \tag{41}$$

Then, the QNMs can be obtained from the "quantization condition"

$$s_n\lambda_{n-1} - s_{n-1}\lambda_n = 0. \tag{42}$$

This is equivalent to giving the iteration number $n$ a truncation. This method is good, but one needs to differentiate the $s(x)$ and $\lambda(x)$ terms for each iteration, which may create problems for numerical precision. To make the process more efficient, Cho et al. [74] improved this method. The improved method does not need to take derivatives at each step, which greatly improves accuracy and speed. What they do is expand the $\lambda_n$ and $s_n$ in a Taylor series around a fixed point $\xi$,

$$\begin{aligned}
\lambda_n(x) &= \sum_{i=0}^{\infty} c_n^i (x - \xi)^i, \\
sn(x) &= \sum_{i=0}^{\infty} d_n^i (x - \xi)^i,
\end{aligned} \tag{43}$$

where $c_n^i$ and $d_n^i$ are the $i$-th Taylor coefficients of $\lambda_n$ and $s_n$, respectively. Using these expressions, one can obtain a set of recursion relations:

$$\begin{aligned}
c_n^i &= (i+1)c_{n-1}^{i+1} + d_{n-1}^i + \sum_{k=0}^{i} c_0^k c_{n-1}^{i-k}, \\
d_n^i &= (i+1)d_{n-1}^{i+1} + \sum_{k=0}^{i} d_0^k c_{n-1}^{i-k}.
\end{aligned} \tag{44}$$

Thus, the "quantization condition" can be re-expressed in terms of the coefficients as

$$d_n^0 c_{n-1}^0 - d_{n-1}^0 c_n^0 = 0. \tag{45}$$

The boundary conditions are pure ingoing waves at the event horizon

$$\psi \sim e^{-i\omega r_*}, \quad r_* \to -\infty, \tag{46}$$

and pure outgoing waves at spatial infinity

$$\psi \sim e^{i\omega r_*}, \quad r_* \to +\infty. \tag{47}$$

It is helpful to transform infinity to be finite, so we perform the following coordinate transformation

$$u = 1 - \frac{r_S}{r}, \tag{48}$$

such that, the range of $u$ is $0 \leq u < 1$. The boundary conditions in terms of $u$ are

$$\psi(u) = \left( \frac{-2(r_S - r_B) + 2\sqrt{r_s - r_B}}{r_S^{5/2} u} \right)^{i\omega r_S^{3/2}/\sqrt{r_S - r_B}} \tag{49}$$

at the horizon, i.e., $u = 0$, and

$$\psi(u) = e^{i\omega(r_S/(1-u) + (r_S + r_B/2))} \left( \frac{4r_S}{1-u} - r_B \right)^{i\omega} \tag{50}$$

at infinity, i.e., $u \to 1$. Thus, we can define

$$\begin{aligned} \psi(u) &= \left( \frac{-2(r_S - r_B) + 2\sqrt{r_s - r_B}}{r_S^{5/2} u} \right)^{i\omega r_S^{3/2}/\sqrt{r_S - r_B}} \\ &\times \left( \frac{4r_S}{1-u} - r_B \right)^{i\omega} e^{i\omega\left( \frac{r_S}{(1-u)} + (r_S + \frac{r_B}{2}) \right)} \chi(u). \end{aligned} \tag{51}$$

Then, the perturbation equations can be rewritten as

$$\chi''(u) = \lambda_0(u)\chi'(u) + s_0(u)\chi \tag{52}$$

where $\lambda_0$ and $s_0$ are functions of $u$ depending on the effective potential. The functions $\lambda_0$ and $s_0$ are complicated so we do not show them explicitly. The first twenty modes for the scalar, electromagnetic, and gravitational fields are shown in Figure 2. The WKB method is powerful in solving frequencies of low-overtone QNMs. We compare the results obtained using the AIM and the WKB method in Tables 1–3 for the scalar, electromagnetic, and gravitational fields, respectively. For low-overtone QNMs, the results obtained using the AIM are in good agreement with those obtained using the WKB method. When the multipole number $l$ increases, the real parts of the QNM frequencies change more noticeably than the imaginary parts, which can be seen in Figure 2 and Tables 1–3. It is important to note that $r_B = 0.5 r_S$ is equivalent to $Q_m = \frac{2\sqrt{2}}{3} \bar{M}$, where $\bar{M} \equiv \frac{\sqrt{3}\kappa_4}{8\pi} M$.

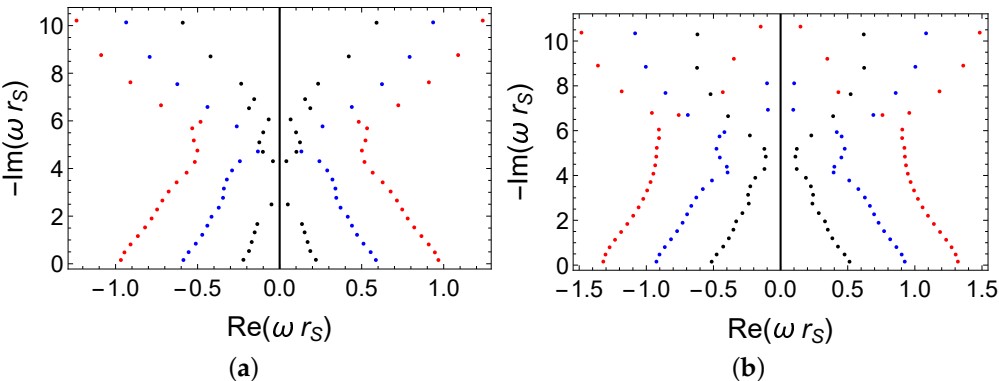

**Figure 2.** *Cont.*

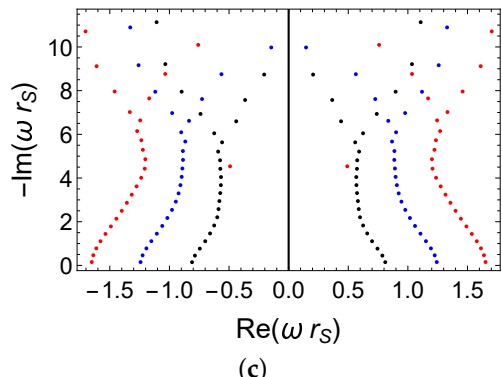

(c)

**Figure 2.** The first twenty QNMs for the charged black hole with scalar hair. The parameter $r_B$ is set to $r_B = 0.5r_S$. (**a**) QNMs for the scalar field with $l = 0$ (black dots), $l = 1$ (blue dots), $l = 2$ (red dots). (**b**) QNMs for the electromagnetic field with $l = 1$ (black dots), $l = 2$ (blue dots), $l = 3$ (the red dots). (**c**) QNMs for the gravitational field with $l = 2$ (black dots), $l = 3$ (blue dots), $l = 4$ (red dots).

**Table 1.** Frequencies of low-overtone QNMs for the scalar field. The parameter $r_B$ is set to $r_B = 0.5r_S$.

| $l$ | $n$ | AIM | | WKB | |
|---|---|---|---|---|---|
| | | $\mathrm{Re}(\omega r_S)$ | $\mathrm{Im}(\omega r_S)$ | $\mathrm{Re}(\omega r_S)$ | $\mathrm{Im}(\omega r_S)$ |
| 0 | 0 | 0.220856 | −0.166592 | 0.219664 | −0.166443 |
| | 1 | 0.186884 | −0.534457 | 0.193004 | −0.537457 |
| 1 | 0 | 0.586476 | −0.158533 | 0.586679 | −0.158525 |
| | 1 | 0.554754 | −0.488295 | 0.556358 | −0.487138 |
| 2 | 0 | 0.967669 | −0.157641 | 0.967666 | −0.157648 |
| | 1 | 0.946096 | −0.478040 | 0.946082 | −0.478076 |

**Table 2.** Frequencies of low-overtone QNMs for the electromagnetic field. The parameter $r_B$ is set to $r_B = 0.5r_S$.

| $l$ | $n$ | AIM | | WKB | |
|---|---|---|---|---|---|
| | | $\mathrm{Re}(\omega r_S)$ | $\mathrm{Im}(\omega r_S)$ | $\mathrm{Re}(\omega r_S)$ | $\mathrm{Im}(\omega r_S)$ |
| 1 | 0 | 0.513377 | −0.152855 | 0.513302 | −0.153142 |
| | 1 | 0.476438 | −0.474101 | 0.476701 | −0.537457 |
| 2 | 0 | 0.924716 | −0.155637 | 0.924714 | −0.155649 |
| | 1 | 0.901934 | −0.472399 | 0.901941 | −0.472447 |
| 3 | 0 | 1.32049 | −0.156374 | 1.32049 | −0.156375 |
| | 1 | 1.30408 | −0.471908 | 1.30408 | −0.471914 |

**Table 3.** Frequencies of low-overtone QNMs for the gravitational field. The parameter $r_B$ is set to $r_B = 0.5r_S$.

| $l$ | $n$ | AIM | | WKB | |
|---|---|---|---|---|---|
| | | $\mathrm{Re}(\omega r_S)$ | $\mathrm{Im}(\omega r_S)$ | $\mathrm{Re}(\omega r_S)$ | $\mathrm{Im}(\omega r_S)$ |
| 2 | 0 | 0.810272 | −0.147554 | 0.810467 | −0.147398 |
| | 1 | 0.785979 | −0.449209 | 0.787063 | −0.447841 |
| 3 | 0 | 1.24218 | −0.152877 | 1.24220 | −0.152875 |
| | 1 | 1.22506 | −0.461665 | 1.22529 | −0.461580 |
| 4 | 0 | 1.65149 | −0.154694 | 1.65149 | −0.154694 |
| | 1 | 1.63833 | −0.465851 | 1.63831 | −0.465856 |

From Equation (13) we know that the mass $M$ and magnetic charge $Q_m$ are closely related to the parameters $r_B$ and $r_S$. When we fix the mass $M$ and increase the magnetic

charge $Q_m$, $r_S$ decreases and $r_B$ increases. So, the effect of the magnetic charge $Q_m$ on the QNMs can be obtained by qualitatively analyzing the effect of the parameter $r_B$ on the QNMs. We study the effect of the parameter $r_B$ on the fundamental QNMs for the three types of perturbed fields. The range of the parameter $r_B$ is $0 \leq r_B \leq 0.5 r_S$, which is equivalent to $0 \leq Q_m \leq \frac{2\sqrt{2}}{3} \bar{M}$. We find that the real parts of the overtone QNMs' frequencies for the scalar and vector fields approximately increase linearly with $r_B$, whereas the absolute value of the imaginary parts approximately decreases linearly with $r_B$, which can be seen in Figure 3a–d. As for the gravitational field, both the real parts and the absolute value of the imaginary parts of the overtone QNM frequencies change slightly with $r_B$ for smaller $r_B$. After a short decrease stage, they then increase rapidly with $r_B$.

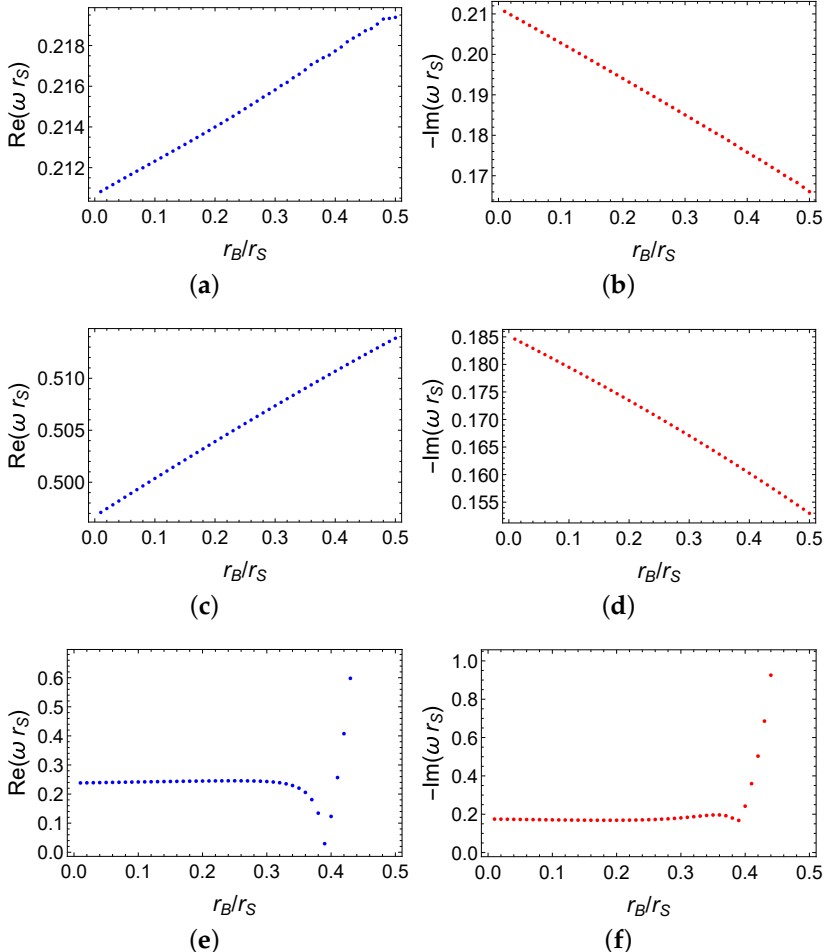

**Figure 3.** The effect of the parameter $r_B$ on the frequencies of fundamental QNMs. (**a**) Real parts of frequencies for the scalar field with $l = 0$. (**b**) Imaginary parts of frequencies for the scalar field with $l = 0$. (**c**) Real parts of frequencies for the electromagnetic field with $l = 1$. (**d**) Imaginary parts of frequencies for the electromagnetic field with $l = 1$. (**e**) Real parts of frequencies for the gravitational field with $l = 2$. (**f**) Imaginary parts of frequencies for the gravitational field with $l = 2$.

*4.2. Time Evolution*

In order to intuitively show the evolution of the perturbed field, we study the QNMs in the time domain, i.e., we do not use the ansatz $\psi \propto e^{-i\omega t}$. Using the null coordinates $u = t - r_*$ and $v = t + r_*$, the perturbation equations can be written in the following form:

$$4 \frac{\partial^2 \Phi_{s,v,g}}{\partial u \partial v} + V_{s,v,g}(r_*) \Phi_{s,v,g} = 0. \tag{53}$$

We choose the initial data as a Gaussian package

$$
\begin{aligned}
\Phi(0, v) &= e^{-\frac{(v-v_c)^2}{2\sigma^2}}, \\
\Phi(u, 0) &= 0.
\end{aligned}
\tag{54}
$$

We choose the package located at $v_c = 10r_S$, with a width $\sigma = 1r_S$. The evolution range is $(0, 200r_S)$, and the results are extracted at $r_* = 20r_S$. The results are shown in Figure 4. By fitting the evolution data, we can also obtain the QNM's frequency. For example, the frequency obtained by fitting the evolution data of the electromagnetic field is $0.512894$–$0.152494i$, which agrees well with the results of the AIM $0.513377$–$0.152855i$. Although the fundamental QNM dominates the evolution of the perturbation, the evolution data are the superpositions of all the QNMs, so this result is good. All three methods obtain the same results, enhancing the credibility of the findings.

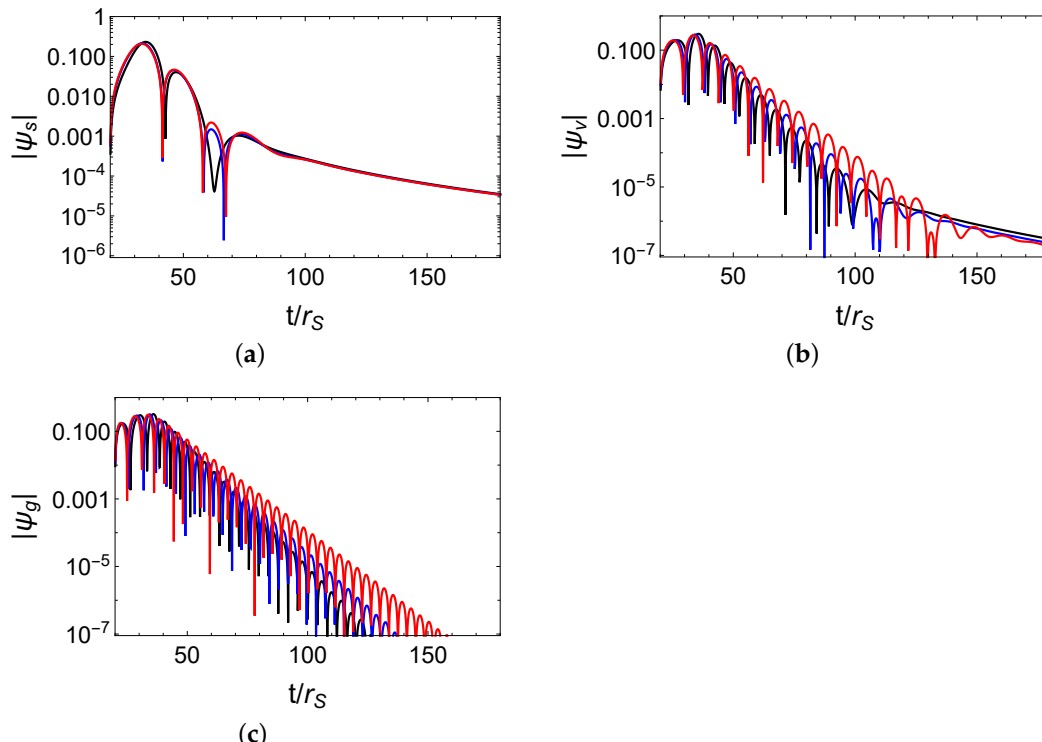

(a)

(b)

(c)

**Figure 4.** Time evolution of the Gauss package extracted at $r_* = 20$. The parameter $r_B$ is set to $r_B = 0.2r_S$ (black lines), $r_B = 0.5r_S$ (blue lines), and $r_B = 0.8r_S$ ( red lines). (**a**) Time evolution of the scalar field with $l = 0$. (**b**) Time evolution of the electromagnetic field with $l = 1$. (**c**) Time evolution of the gravitational field with $l = 2$.

## 5. Conclusions

Through the KK reduction, the five-dimensional Einstein–Maxwell theory reduces to a four-dimensional Einstein–Maxwell dilaton theory, which supports a spherically static charged black hole solution. We studied the linear perturbation equations of the scalar, electromagnetic, and gravitational fields in the background of this spherically static charged black hole. Because of the spherical symmetry of the background, the radial parts of the perturbed fields can be decomposed from the angular parts. Using the tortoise coordinate $r_*$, every perturbation equation can be written in a Schrödinger-like form. The effective potentials for the scalar, electromagnetic, and gravitational fields are shown in Figure 1. In this figure, we can see that the effective potentials, except for that of the electromagnetic field, depend on the parameter $r_B$.

Using the AIM, we computed the QNM frequencies for the three types of perturbed fields. As the multipole number $l$ increased, the real parts of the QNM frequencies changed

more noticeably than the imaginary parts, which can be seen in Figure 2 and Tables 1–3. We also compared the results obtained using the AIM and the WKB method and found that for low-overtone QNMs, the results obtained using the AIM were in good agreement with those obtained using the WKB method. The effect of the parameter $r_B$ was also studied. Using the null coordinates $u$ and $v$, the evolution of a Gaussian package was also investigated. The results showed that the QNM frequencies obtained by fitting the evolution data agreed well with the results obtained using the AIM.

It is important to note that we only studied the QNMs for a charged black hole. For the topological star, there is no event horizon so the ingoing boundary condition cannot be imposed. Thus, it should be treated separately. We will study this in the future.

**Author Contributions:** W.-D.G. did the calculation and wrote the draft of the paper, Q.T. offered the method for solving the QNFs, discussed the details and improved the draft. All authors have read and agreed to the published version of the manuscript.

**Funding:** This work was supported by the National Key Research and Development Program of China (Grant No. 2020YFC2201503), the National Natural Science Foundation of China (Grant Nos. 12205129, 12147166, 11875151, 12075103, and 12247101), the China Postdoctoral Science Foundation (Grant No. 2021M701529), the 111 Project (Grant No. B20063), and Lanzhou City's scientific research funding subsidy to Lanzhou University.

**Data Availability Statement:** Data available on request from the authors.

**Acknowledgments:** We thank the referees'suggestions for improving this paper, and we thank the editor's effort and time during the publication process.

**Conflicts of Interest:** No potential conflict of interest was reported by the authors.

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
