# Peer review of "Quasinormal Modes of a Charged Black Hole with Scalar Hair"

_universe, doi:10.3390/universe9070320_

Round 1
Reviewer 1 Report
The authors consider the quasinormal modes of the magnetically charged static black holes, which are obtained by the Kaluza-Klein reduction, in the four-dimensional Einstein-Maxwell-dilaton theory. Using the asymptotic iteration method and the Wentzel-Kramers-Brillouin method, the authors perturbatively study the effects of the magnetic charge on the quasinormal modes qualitatively. I find this work interesting but I recommend the authors to consider the following points:
i) What is the astrophysical motivation for including the magnetic charges in the present study of the black hole quasinormal modes? For example, would such a magnetic charge increase the detectability of the quasinormal modes in the present and the future astrophysical and astronomical observations? Or, would one obtain the astrophysical upper limits of the magnetic charge by some observations? Then the authors may quantitatively discuss such properties of the magnetic charge in this study in detail.
ii) Similar to the papers arXiv:2006.08957 and arXiv:2206.07721, would the master equations given in the section III be solved analytically by some Heun functions? If so, the authors may add similar calculations in the current geometry and compare their present results with the Heun function methods.
iii) Why does the effective potential of the electromagnetic perturbation not depend on the magnetic charge? The authors may add some reasons or predictions.
Author Response
Dear Editor, Thanks for your very valuable comments and suggestions. We have updated the manuscript. The changes are marked in blue color. Replies to the comments are listed as follows. Comment of Report of Referee and our responses: The authors consider the quasinormal modes of the magnetically charged static black holes, which are obtained by the Kaluza-Klein reduction, in the four-dimensional Einstein-Maxwell-dilaton theory. Using the asymptotic iteration method and the Wentzel-Kramers-Brillouin method, the authors perturbatively study the effects of the magnetic charge on the quasinormal modes qualitatively. I find this work interesting but I recommend the authors to consider the following points: Comment i) What is the astrophysical motivation for including the magnetic charges in the present study of the black hole quasinormal modes? For example, would such a magnetic charge increase the detectability of the quasinormal modes in the present and the future astrophysical and astronomical observations? Or, would one obtain the astrophysical upper limits of the magnetic charge by some observations? Then the authors may quantitatively discuss such properties of the magnetic charge in this study in detail. Reply: The main effect of this magnetic charge is to support the topological star and to avoid the instability. As for the astrophysical motivation, the only reason is that we can not exclude the magnetic charge from observation. And we may discuss the effect of the magnetic charge in the future. Comment ii) Similar to the papers arXiv:2006.08957 and arXiv:2206.07721, would the master equations given in the section III be solved analytically by some Heun functions? If so, the authors may add similar calculations in the current geometry and compare their present results with the Heun function methods. Reply: We thank the referee for pointing out this issue. We can’t solve the master equations by some Heun functions. But we have cited these two papers in the manuscript of the new version. Comment iii) Why does the effective potential of the electromagnetic perturbation not depend on the magnetic charge? The authors may add some reasons or predictions. Reply: Actually, the effective potential of the electromagnetic perturbation depends on the magnetic charge. From Eq. (26) we know that the effective potential of the electromagnetic perturbation depends on the parameter . Besides, from Eq. (13) we know that is related to the magnetic charge. So the effective potential of the electromagnetic perturbation depends on the magnetic charge. We have added this at the end of Sec.III B in the manuscript of the new version. Thank the referee and editor again. Best regards, Wen-Di Guo

Reviewer 2 Report
Please, see the attached file.

Author Response
Dear Editor and Referee,
\\
\\
Thanks for your very valuable comments and suggestions. We have updated the manuscript. The changes are marked in blue color. Replies to the comments are listed as follows.
\\
\\
\textbf{Comment:} The authors study the potentials for scalar, electromagnetic and gravitational perturbations and the time evolution of QNM modes in a 4D hairy charged black hole obtained after the reduction of the 5D KK theory. The authors also
compared the frequencies from the AIM and WKB methods and they are shown to yield reasonably similar results.
However, the authors should clarify two things:
1. Define clearly what is Ry in Eq.(8). Is it the compactified 5th dimension over radius $R$?
\\
\textbf{Reply:} Thank the referee for pointing out this. Yes, it is the radius of the extra dimension. We have added it below Eq.~(8).
\\
\\
\textbf{Comment:} 2. I cannot …nd the ADM mass M as in Eq.(12). Please check if the conformal factor $f_B^{1/2}$ in Eq.(6) is correctly typed. If it’s correct, then I get using the usual integral for ADM mass the value $M = \frac{r_S}{2}$. If instead the conformal factor is $e^{2\Phi}$ or $f_B^{-1/2}$ in Eq.(6), I get $M = r_B + \frac{r_S}{2}$ . Which expression for ADM
mass is the correct one?
I recommend that the paper be published in the Universe after the above clarifications are introduced.
\\
\textbf{Reply:} The result in our manuscript is correct. Here I list the derivation of the ADM mass as follows:
In an asymptotically flat spacetime, we can write the metric as
\begin{equation}
g_{\mu\nu}=\eta_{\mu\nu}+h_{\mu\nu}.
\end{equation}
Here, we only focus on the spatial components. The definition of the ADM mass is
\begin{equation}
M_{ADM}=\frac{1}{2\kappa_4}\int_{\partial\Sigma}d^2x\sqrt{\gamma^{(2)}}\sigma^i(\partial_i h^j_i-\partial_i h^j_j),
\end{equation}
where $\partial\Sigma$ is the two-sphere at spatial infinity, $\sqrt{\gamma^{(2)}}$ is the determinant of the induced metric on the two-sphere, and $\sigma^i$ is the outward-pointing normal vector, in the $(r, \theta, \phi)$ coordinate it is $(1,0,0)$. Taking $t=constant$ as the hypersurface $\Sigma$, the induced metric can be written as
\begin{eqnarray}
ds^2&=&f_B^{-1/2}f_S^{-1}dr^2+f_B^{1/2}\left(r^2 d\theta^2+r^2\sin^2\theta d\phi^2\right)\nonumber\\
&=&f_B^{-1/2}f_S^{-1}\left[dr^2+f_Bf_S(r^2 d\theta^2+r^2\sin^2\theta d\phi^2)\right]\nonumber\\
&=&(1-r_B/r)^{-1/2}(r-r_S/r)\left[dr^2+(1-r_B/r)(r-r_S/r)(r^2 d\theta^2+r^2\sin^2\theta d\phi^2)\right]\nonumber\\
&\approx&(1+\frac{1}{2}\frac{r_B}{r}+\frac{r_S}{r})\left[dr^2+r^2 d\theta^2+r^2\sin^2\theta d\phi^2\right].
\end{eqnarray}
We have $h_{ij}=\text{diag}(\frac{1}{2}\frac{r_B}{r}+\frac{r_S}{r},r^2(\frac{1}{2}\frac{r_B}{r}+\frac{r_S}{r}),r^2\sin^2\theta(\frac{1}{2}\frac{r_B}{r}+\frac{r_S}{r}))$. Substituting this into the definition of ADM mass, we have
\begin{eqnarray}
M_{ADM}&=&\frac{1}{2\kappa_4}\int r^2\sin\theta(\partial_r h^r_r-\partial_r h^j_j)d\theta d\phi\nonumber\\
&=&\frac{1}{2\kappa_4}\int r^2\sin\theta\left(-\frac{\frac{1}{2}r_B+r_S}{r^2}-3\left(-\frac{\frac{1}{2}r_B+r_S}{r^2}\right)\right)d\theta d\phi\nonumber\\
&=&\frac{1}{2\kappa_4}(r_B+2r_S)\int \sin\theta d\theta d\phi\nonumber\\
&=&\frac{1}{2\kappa_4}(r_B+2r_S)4\pi\nonumber\\
&=&2\pi\left(\frac{r_B+2r_S}{\kappa_4}\right).
\end{eqnarray}
This exactly is the result in our manuscript.
Thank you for the very useful comments again.
Best regards,
Wen-Di Guo

Round 2
Reviewer 1 Report
I am satisfied with almost all of the authors' corrections. However, some recent studies [1-3] consider some astrophysical and astronomical effects of magnetic charges in some black hole spacetimes. Then the present authors may cite these papers [1-3] and add their motivation to study the magnetically charged black holes.
[1] D.~Ghosh, A.~Thalapillil and F.~Ullah, ``Astrophysical hints for magnetic black holes,'' arXiv:2009.03363 [hep-ph].
[2] M.~D.~Diamond and D.~E.~Kaplan, ``Constraints on relic magnetic black holes,'' arXiv:2103.01850 [hep-ph].
[3] V.~Karas and Z.~Stuchlik, ``Magnetized black holes: interplay between charge and rotation,'' arXiv:2306.07804 [gr-qc].
Author Response
Dear Editor and Referee,
Thanks for your very valuable comments and suggestions. We have updated the manuscript. The changes are marked in blue color. Replies to the comments are listed as follows.
Comment: I am satisfied with almost all of the authors' corrections. However, some recent studies [1-3] consider some astrophysical and astronomical effects of magnetic charges in some black hole spacetimes. Then the present authors may cite these papers [1-3] and add their motivation to study the magnetically charged black holes.
[1] D.~Ghosh, A.~Thalapillil and F.~Ullah, ``Astrophysical hints for magnetic black holes,'' arXiv:2009.03363 [hep-ph].
[2] M.~D.~Diamond and D.~E.~Kaplan, ``Constraints on relic magnetic black holes,'' arXiv:2103.01850 [hep-ph].
[3] V.~Karas and Z.~Stuchlik, ``Magnetized black holes: interplay between charge and rotation,'' arXiv:2306.07804 [gr-qc].
Reply: Thank the referee for pointing out these three papers about the astronomical effects of magnetic charges. It is helpful to improve our manuscript. We have added them at the end of Sec.II in the manuscript of the new version.
Thank you for the very useful comment again.
Best regards,
Wen-Di Guo
